# Who presents the greatest challenge in intellectual disability research- participants or health and research professionals?

Rosemary Kelly[1]*, Laurence Taggart[1], Vivien Coates[2], Maria Truesdale[3], Alison Dunkley[4], Michelle Hadjiconstantinou[4], Kamlesh Khunti[4], Nicola Mills[5]

1 School of Nursing and Midwifery, Queen's University Belfast, United Kingdom, 2 School of Nursing & Paramedic Science, Ulster University Belfast, United Kingdom, 3 College of Medical Veterinary and Life Sciences, University of Glasgow, United Kingdom, 4 Leicester Diabetes Research Centre, University of Leicester, United Kingdom, 5 Bristol Medical School, University of Bristol, United Kingdom

* rosie.kelly@qub.ac.uk

## Abstract

### Background

The challenges of recruitment to randomised controlled trials have been well documented. The additional challenges of recruiting people with intellectual disabilities (ID) and significant health co-morbidities have been the focus of less attention. The aim of this work was to explore issues around the screening and recruitment of adults with ID and Type 2 Diabetes (T2D) into the internal pilot of the 'My Diabetes and Me' Randomised Controlled Trial. The findings were used to develop recommendations and implement interventions to address challenges for recruitment to the main study.

### Methods

A multiple methods approach using the QuinteT Recruitment Intervention was employed across three National Health Service sites in the United Kingdom. Semi-structured interviews were undertaken with staff, and adults with ID/T2D recruited to the study; analysis of recruitment discussion recordings, and a review of documentation pertaining to screening logs and research meetings was also performed. Thematic analysis identified the complexity of challenges and potential enablers to recruitment in this population.

### Results

Recruitment challenges began much earlier than anticipated with significant organisational process challenges to be overcome. The discomfort felt by some staff in putting potential participants forward was evident as they don't feel the study is appropriate for this population, or they feel out of their depth. Engagement and

**Data availability statement:** While this was not a requirement of the ethical approval granted for this study, the authors of this paper are of the opinion that it is not appropriate to share the data from this study openly as it is a small sample. While it is de-identified, it is potentially possible that some of the participants could be identified through the quotes and descriptions contained within the data. One of the criteria for participant consent was that they would not be identifiable in any publications so we must adhere to that principle. To that end I can confirm that the following individual has agreed to be the non-author contact for the School of Nursing & Midwifery at Queen's University, Belfast to which data requests may be sent: Professor Mark Linden School of Nursing and Midwifery Room 05.308 – Medical Biology Centre Queen's University, Belfast BT7 1NN, United Kingdom Email: m.linden@qub.ac.uk Phone: +44 (0)28 9097 2820.

**Funding:** Award made to Professor Laurence Taggart (LT), Ulster University Reference No: COM/5637/20 www.research.hscni.net The sponsors did not play any role in the study design, data collection, analysis, decision to publish or preparation of the manuscript.

**Competing interests:** No authors have competing interests.

'buy-in' from people with ID/ T2D were much more positive. The emotional labour involved in ensuring the internal pilot progressed to the main study was noteworthy.

## Conclusion

The use of QRI methodology within an ID RCT is a novel approach, unearthing emotional challenges and significant systemic organisational process challenges. The findings of this study clearly illustrate the informed implementation strategies that are needed to improve recruitment processes, minimise the emotional labour relating to NHS organisational 'buy-in' and address the preparation and readiness of NHS health and social care staff for research in under-represented populations.

## Introduction

Recruitment to randomised controlled trials (RCTs) is challenging and failure to recruit the required number of participants is the main reason for trials discontinuing [1–2]. Recruitment is not a single process, but involves screening, identifying, and approaching potential participants and there can be bottlenecks at any point in this pathway prior to consent that can hinder meeting recruitment targets. Recruiting participants from vulnerable or under-represented populations such as people with intellectual disabilities (ID) may pose particular ethical and methodological challenges, that will subsequently require reasonable adjustment to RCT approaches [3–4]. Current health and social care research policy in the United Kingdom (UK) advocates for the inclusion of patients, service users and the public in all aspects of research, including design, conduct and dissemination [5]. While there is an increasing body of work exploring engagement with the ID population around participation in the research process generally [6–7], and in RCTs specifically [8], significant challenges remain related to the individual with ID, the RCT methodology and system issues [9].

### Individuals with intellectual disability

Individuals with ID often present with communication and cognitive functioning challenges, which may make obtaining informed consent more problematical [4]. An added layer of complexity for people with ID, is that they are likely to have co-morbidities which may increase their physical vulnerability, and, with a lack of insight into the prevalence of health conditions in this population [10–11], access may be confined to smaller numbers of eligible participants [12].

Researchers often experience more complex engagements with family members and National Health Service (NHS) health and social care teams when including people with ID in research studies [3]. As a result, decisions about inclusion of these individuals may be made by multiple people at multiple time points, for example, by NHS health and social care staff or by study team members (when considering screening and eligibility criteria), or by other individuals with a caring role making paternalistic decisions based on their perception of the potential trial participant's competence/ willingness to participate.

## RCT methodology

There are practical issues around accessible research design and adapting resources, such as the participant information sheet and consent form [13], and the need for multiple consent procedures [14], which may lead RCT researchers to intentionally or unintentionally exclude people with ID from their studies [15]. Such exclusion may deny this group the benefits from improvements in treatment which will be subsequently available to the general population [16].

In recognition of the challenges around recruitment for specific populations [17–18], tools such as PRECIS-2 have been generated to help researchers with their design decisions, to give greater focus and support to addressing such issues [19]. However, despite growing recognition of the complexity related to recruitment into RCTs, and particularly for specific populations, there remains a lack of effective solutions [20]. While these issues have been raised before, how they influence the decision-making of people responsible for identifying and recruiting people with ID has rarely been considered [21].

## NHS health and social care systems

The complexity of including people with ID in RCTs, as described above, can make it difficult for researchers to secure adequate funding to carry out trials, despite the development of the UK Research Framework by the Health Research Authority (HRA) [5] which supports inclusivity. It can also be challenging to get buy-in from NHS health and social care organisations across the UK who have limited workforce resources to support individuals with ID, adding another layer of complexity to accessing this population [22]. The involvement of disparate organisations across health, social care and academia inevitably leads to a certain amount of bureaucracy and can result in delays within research governance, finance, and staff recruitment processes prior to the commencement of the identification and recruitment of potential participants [23].

An added complication has been the impact of the COVID-19 pandemic on healthcare in general and on the ability of organisations to prioritise and support research currently [24]. The impact of limited routine healthcare for people with ID during the pandemic has been identified [25], however, the current impetus to address significant backlogs in healthcare delivery for the public, coupled with the aforementioned challenges conducting RCTs with people with ID, suggests that anecdotally, research priorities are less likely to focus on this smaller cohort.

Type 2 Diabetes (T2D) is reported to be 2–3 times higher in adults with ID than in the general population [26–27], due to certain chromosomal conditions (i.e. Down's Syndrome) increasing the risk of developing T2D. People with ID are also more likely to lead unhealthier lifestyles that include poor diets, decreased physical activity, increased sedentary behaviour and greater use of antipsychotic medication leading to higher levels of abdominal obesity [28]. Furthermore, people with ID develop T2D at an earlier age making management protracted and more complex [29].

Previous studies have identified initiatives designed to promote healthier outcomes for people with ID/T2D [29–30] and the current study is an RCT using a specifically designed programme for this population.

## The my diabetes and me study exemplar

The structured diabetes self-management education programme – DESMOND runs successfully via the NHS in the UK and is available to the general population who have been diagnosed with T2D [31]. It has not, however, been accessible to people with ID/T2D [11,26]. An earlier pilot feasibility study demonstrated that a modified structured self-management education programme known as DESMOND-ID was acceptable and can be successfully delivered to adults with ID/T2D [32]. The National Institute for Health and Care Research (NIHR) has recently funded a multi-centre RCT with an internal pilot to test the efficacy and impact of the DESMOND-ID intervention in study sites across the UK (My Diabetes and Me Study – HTA NIHR 131692). Three sites participated in the internal pilot study with a further eight sites participating in the full study. Participants were randomised to the Intervention Group (DESMOND-ID Structured Education Programme) or

the Control Group (education booklet and treatment as usual). The earlier pilot feasibility study identified that the identification and recruitment of adults with ID/T2D was more challenging than anticipated [29]. With this in mind, an established intervention – the QuinteT Recruitment Intervention (QRI) – was embedded within the internal pilot of the NIHR definitive study to focus on and address the complexities and barriers of recruiting adults with ID. The aim of this QRI study was to understand and address the enablers and barriers in how clinical staff identified and approached adults with ID/T2D to participate in an RCT. In this paper, we report on those enablers and barriers using the 'My Diabetes and Me' Study as an exemplar.

## Materials and methods

### The QuinteT Recruitment Intervention within the internal pilot of the 'My Diabetes and Me' Study

COREQ (COnsolidated criteria for REporting Qualitative research) Checklist is included in S1 File.

The purpose of the QRI was to identify and address issues that were hindering effective recruitment [33]. Table 1 illustrates the QRI – supported triangulation of multiple data sources including:

- interviews with key stakeholders.

- audio-recording of recruitment-to-study discussions.

- detailed screening logs using the SEAR framework [34].

**Table 1. Data collection methods.**

| Method | Detail |
|---|---|
| a) Recorded interviews with: | |
| 1. HSC staff who worked with individuals who had ID/T2D, and researchers involved in the study | Staff were interviewed face to face at their place of employment or remotely using MS Teams™. All interviews were audio- recorded, transcribed verbatim and de- identified prior to analysis. |
| 2. Senior leaders from Clinical Research Networks aligned to the study | |
| 3. People with ID/T2D recruited to the study | People with ID/T2D were interviewed face to face in their homes or remotely via MS Teams™. All participants were invited to have someone sit with them if they wished during the interview. Some chose to have staff members nearby but not directly involved in the interview. Interviews were audio-recorded, transcribed verbatim and de- identified prior to analysis. |
| b) Mapping of study recruitment pathway and scrutiny of screening logs. | To compare and contrast the different approaches undertaken to recruit study participants, – and to look for patterns within and across sites with regard to the number of people identified, approached and consented to the study. |
| c) Audio-recordings of recruitment-to-study discussions between study researchers and eligible study participants. | To assess how information about the study was conveyed and how people responded to it. |
| d) Review of study documentation such as easy-read PIS and invitation letters | To ensure key study documents were clear and balanced (e.g., not biased towards or against the intervention). |
| e) Attendance at/review of minutes from trial management meetings and investigator meetings. | To gain an overview of study conduct and overarching challenges. |

- mapped recruitment pathway.

- review of study documentation (easy-read Participant Information Sheets (PIS) and invitation letters) and minutes of Trial Management Group and Investigator meetings.

Recruitment of staff began on 21st October 2022, with data collection beginning on 4 November 2022. Given the delays in appointing staff into post which will be discussed, recruitment and data collection of staff continued until 29th September 2023. Staff were identified through their involvement in the RCT and a snowballing process via management and research teams. Recruitment of people with ID began on 1st April 2023 with data collected from 26th April – 15th August 2023. Individuals were contacted when they had been randomised to the study. These data were used to investigate how the study recruitment process worked (enablers) in each study site, and to elucidate key bottlenecks (barriers). Findings were synthesized and discussed with the study's Chief Investigator and trial management group (TMG), and actions formulated and implemented to optimise recruitment in the pilot stage, or to offer recommendations for the full RCT.

Semi-structured interviews were undertaken by the first author (RK) with NHS health and social care staff involved in the identification of potential participants with ID/T2D, either through their professional role, or as leaders within NHS services across three sites in the UK. These included community ID nursing teams, diabetes nurse specialists, GPs, and practice nurses. RK is a registered nurse with a PhD. This author was the lone data collector with additional responsibility for data curation, development and refinement of aspects of the documentation as it applied to this work, e.g., screening logs, topic guides etc., project administration, transcription and coding in partnership with other authors and writing all manuscript drafts.

Research Associates and Clinical Research Nurses (CRNs) employed for recruitment and data collection duties along with senior leaders employed in clinical research networks related to the UK sites were also interviewed in February/ March 2023 to provide insight into some of the organisational challenges related to multi-site research that had been identified in earlier interviews. People with ID/T2D who had consented to randomisation within the internal pilot were also invited to take part in an interview with the first author (RK) to explore their views on the study and the recruitment process. Topic guides were developed drawing upon findings from the feasibility study [32] and previous QRI studies [35]. These were revised as data collection and analysis progressed to advance developing themes from earlier interviews as detailed in S2 and S3 Files.

Participants were not asked to review their transcripts. A meeting was held after the initial data analysis had been completed, between the research team and many of the professionals interviewed. Early findings were discussed, and further insight was added to the data collection to enhance the coding process (Fig 1). People with ID who took part in the interviews were not sent their transcripts for review, but an easy-read summary has been produced as part of the dissemination process.

## Data analysis

Data from the interviews and recruitment discussions were subjected to full transcription and de-identification by the first author (RK). All transcriptions were pseudoanonymised using a description of role/site/interview order. Interviews with staff lasted 35–50 minutes while interviews with people with ID/T2D lasted 9–20 minutes. A code book detailed in S4 File was developed based on both inductive and deductive thematic analysis which was regularly shared and discussed with the team [36]. A theoretical framework was not used for this study as the team wished to approach the study with an open mind and with equipoise, looking at the data independently and collectively, with regular discussions as codes were added to and refined as more data were collected. This is in keeping with the pragmatic approach adopted during the initial development of the QRI approach, which used multiple methodological underpinnings [33].

Further independent analysis on a selection of interview transcriptions was carried out by VC, and recruitment discussions were analysed independently by NM and discussed amongst the research team to ensure trustworthiness of

the process. Documentation analysis by the first author (RK) included screening logs, recruitment pathways at each site, and agreed minutes of all meetings. Issues arising from the documentation were added to the existing codes, with new codes added as needed. The findings relating to aspects of recruitment which appeared to be working well and those areas working less well were shared with sites to raise awareness and so that by working together, challenges could be addressed for the internal pilot and main RCT. Our experience of using the QRI methodology will be reported in a separate paper.

## Ethics approval

The QRI was integrated after ethical approval for the internal pilot stage of the RCT had already been granted. Separate UK Research Ethics Committee approval was therefore sought and approved for the QRI component (22/HSC/0009).

Informed consent was obtained from all participants. Health and Social Care and research staff were provided with a Participant Information Sheet (PIS) detailing the study and a consent form which they were asked to sign and either return electronically to RK or sign on the day if the interview was face to face. Information about the QRI study was included in the easy-read format PIS provided to people with ID/T2D about the internal pilot study. The QRI interview was included in the recruitment discussion about the internal pilot study and written consent obtained. RK confirmed their continuing consent when the participants were contacted to arrange the interview, and again on the day of the interview.

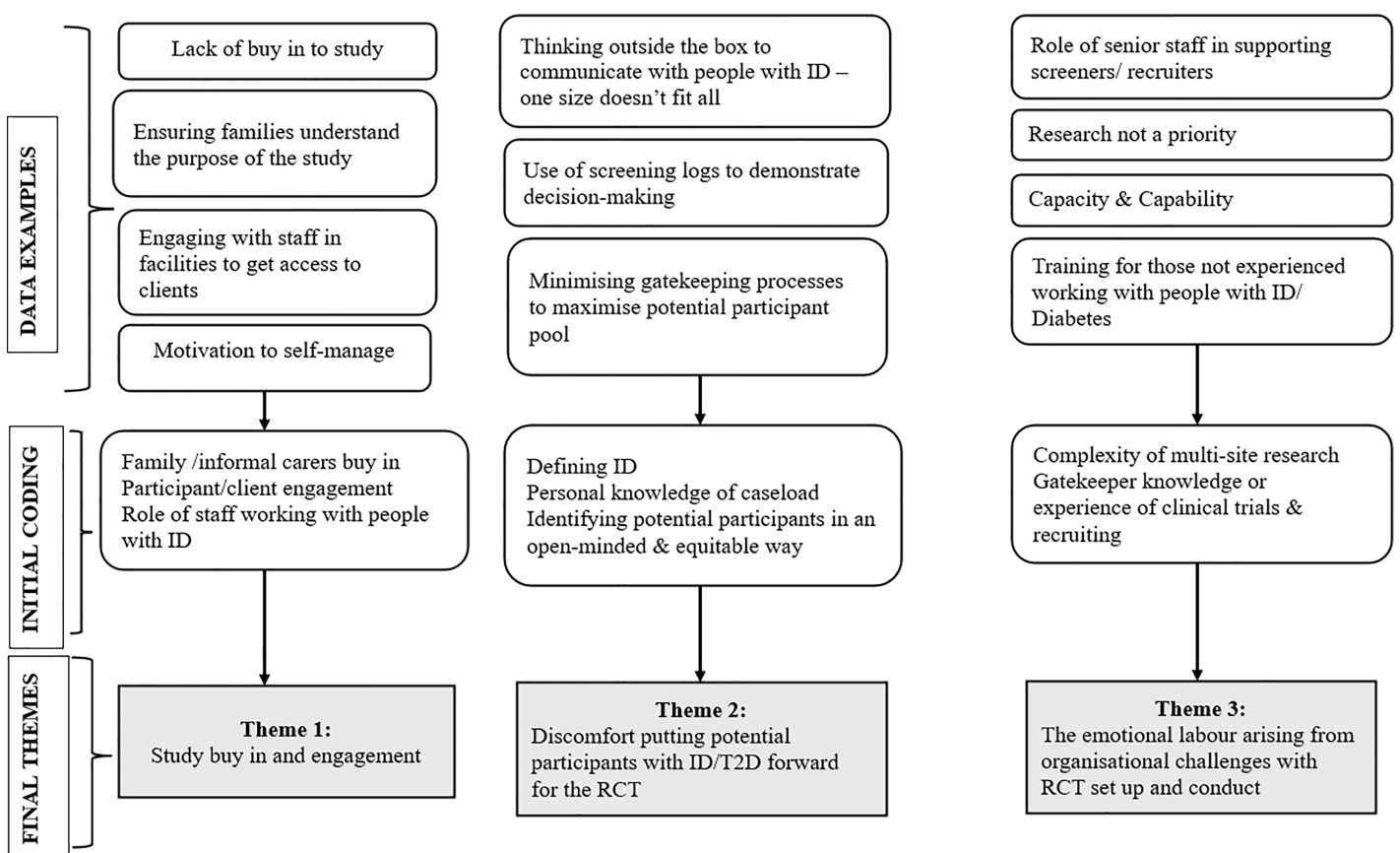

**Fig 1. Coding tree.**

# Results

## Data sources

Interviewees detailed in Table 2 included:

- NHS health and social care staff including senior managers, clinical network staff and ID team members (n = 14)
- Research staff (n= 6)
- People with ID/T2D who were recruited to the study (n=5)

• Analysis of:

- Recruitment discussions (n=6 from 1 site)
- Screening logs from 1 site (n=20)
- Minutes of study meetings (n=27)
- Activity log of communication involved in coordinating the internal pilot and arranging the interviews (n = 1).

## Thematic analysis of the core recruitment challenges

Fig 1 illustrates a coding tree was developed as art of the analysis process to illustrate the initial theme development and subsequent revisions until the final themes were agreed. The thematic analysis identified a number of themes which were aggregated into three core recruitment challenges: (i) study buy-in and engagement; (ii) discomfort putting potential participants with ID/T2D forward for the RCT; and (iii) the emotional labour arising from organisational challenges with RCT set up and conduct.

Interview data was the main source of evidence used to develop the themes, with other sources of data such as screening logs and minutes of meetings examined to support what was being said in the interviews or to offer additional insights into the management of issues as they arose. Only one NHS site returned screening log evidence illustrating that although a large proportion of patients were screened for eligibility (n = 187) and 142 were deemed eligible, only 65 were approached so they could consider taking part and 39 were subsequently randomized (unsuitable n = 13; declined n = 13). Data from this log, and interviews with NHS health and social care staff from the other two sites indicated that most people were excluded early in the recruitment pathway as they were considered not suitable for the study (even if they met the eligibility criteria as per protocol) or they, or their carer, did not wish to meet to hear more about the study. Those that were happy to meet with a researcher to discuss the study tended to consent to participation (75%).

## Theme 1: Study buy-in and engagement

**Mixed opinions on empowering people with ID/T2D.** The interviews revealed mixed opinions on empowering people with ID/T2D to self-manage their long-term health conditions and how this might be achieved. At one end of the scale, NHS health and social care staff and researchers who worked in the ID specialism reported that people with

**Table 2. Interview characteristics.**

| | Professional Role | Research staff | Clinical staff | Managers | Adults with ID/T2D |
|---|---|---|---|---|---|
| Site 1 | Nursing, medical practitioner, research & development staff | 2 males 2 females | 6 females | 1 male | 4 males |
| Site 2 | Nursing, medical practitioner, research & development staff | 1 female | 1 male | | |
| Site 3 | Nursing, medical practitioner, social care practitioners, research & development staff | 4 females | 1 female | 2 females | 1 female |

disabilities should be offered the same opportunities to avail of evidence-based self-management interventions (such as DESMOND-ID) in a way that met their needs, as people without disabilities. At the other end of the scale, there were views that self-management interventions for those with T2D would be less useful for people with ID because they were thought to be either managing their condition well or were dependent on their carers. These opinions influenced how staff viewed the study and their subsequent buy-in or lack of buy-in with it:

*"…My view is people with disabilities should be entitled to the same opportunities in a way that meets their needs, so if DESMOND is shown to be very useful in reducing and helping people with Type 2 Diabetes then certainly, it should… there should be an attempt for working with people with a disability."* (ID nurse/researcher 01002)

*"…so not saying it won't be a useful intervention but there's not many people I know Who are managing, nobody I know with a learning disability over the years who has Really managed their Type 2 diabetes. The people who are managing it wouldn't need the additional support [from the study intervention] and the people who aren't are dependent on their support team."* (HSC clinician 03001)

**Complexities of engaging multi-specialty teams.** Researchers in the study had to work closely with different ID organisations to identify potential participants given that services for people with ID and T2D can be accessed through multiple pathways in the community, as shown in Fig 2.

While all the NHS sites identified multiple teams who could assist with identification and recruitment, for example supported living and residential facilities, day centres, and voluntary support organisations, community ID teams were

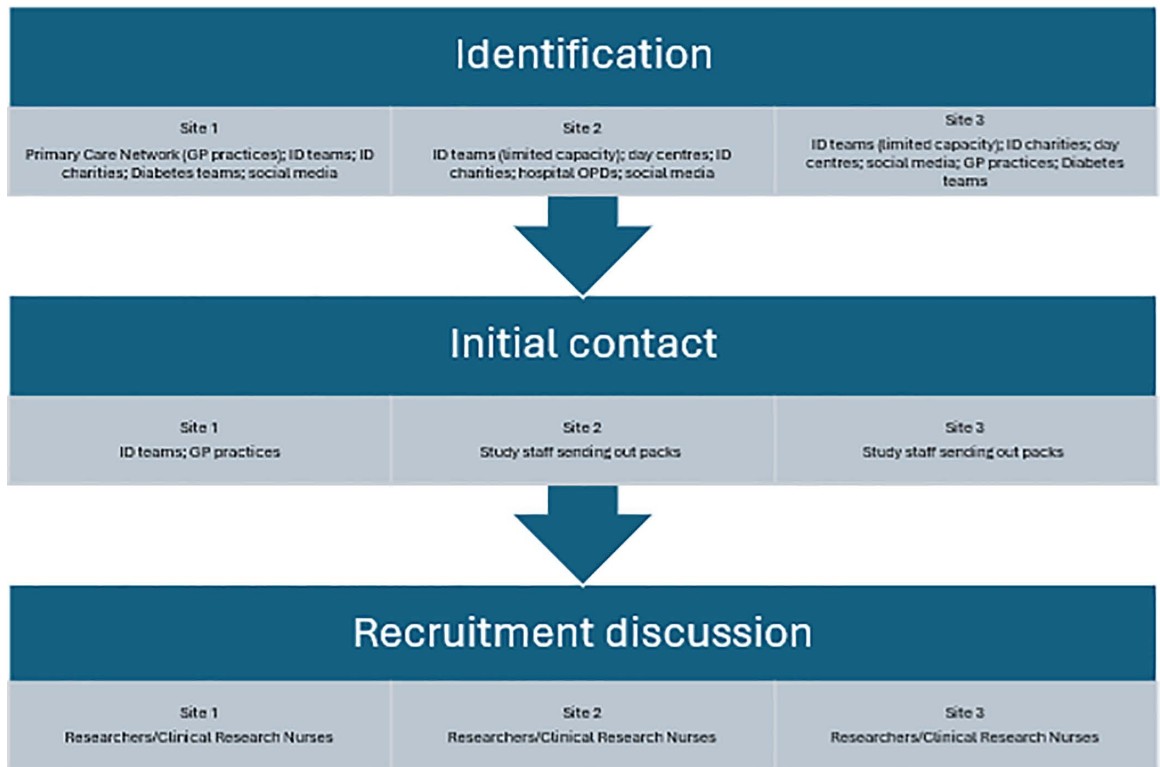

**Fig 2. Recruitment pathway and the multidisciplinary teams involved.**

considered to be the richest source to identify potential study participants, given their expertise in working with people with ID. By starting study preparation in advance of initiating the recruitment process, Site 1, who reached their recruitment target and recruited the majority of the total number of consented adults with ID/T2D in the internal pilot, had community ID teams engaged with initial screening, and the research team reported an effective collaborative relationship with them. As the study progressed, it was noticeable that NHS sites 2 and 3 who were less successful in terms of recruitment, had less time for study preparation due to bureaucracy delays, resulting in struggles to engage community ID teams in the screening process and those who used clinical research network facilities to conduct the recruitment discussions. It was also noted that screening by other services at site 1 and reports at meetings from sites 2 and 3 illustrated that staff with less experience of working with this ID population were more inclined to exclude greater numbers of potential participants at either the initial screening stage or at the recruitment discussion stage, and had a less successful collaborative working relationship with the site research team:

> *"But whenever it comes down to it, I've been chasing, chasing, chasing, you know, promise me, you're gonna generate a list of potential participants. And that's been quite a struggle because it's like, follow up telephone conversations, phone calls, which have been returned back to me follow up emails, follow up, but nothing has come from us, which is disappointing given the initial meeting."* (Researcher 03002)

What was underestimated as the pilot study progressed was how much engagement the research team needed to have with NHS health and social care teams to reinforce the potential benefits of the study, before researchers ever got to speak to potential participants with ID. It became evident that fostering a strong relationship between NHS health and social care ID teams and research teams was required to enable a good understanding of the study and its rationale, and to support buy-in:

> *"…So, whenever you do research it's really, really important that you have a very strong relationship with your clinical partners way before the intervention starts and get their buy-in. So they would have ownership as well of the project. I think it's really difficult just to rock up to someone and say 'hey, do you want to be involved in a trial?' Understanding the trial, understanding all the processes has been very, very complicated.*" (Researcher 01007)

**Staff anticipated family/carer engagement issues.** Concern was expressed by NHS health care staff around family/paid carer commitment and knowledge about diabetes that might act as a barrier to study participation. Some NHS health and social care staff expressed concern that engagement of family/paid carers to the study would prove challenging because they would not be willing or able to provide the additional level of support needed to ensure participants adhered with all the elements of the study, or that they appreciated the importance of supporting a healthier lifestyle for this population. This created a degree of reluctance by some staff to facilitate the engagement of potential participants and their carers with the study:

> *"…So then you're looking at people and whether or not they live in supported living, you're looking at a member of staff being trained to support them to attend or a family member being very committed to bring them. I suppose that's a barrier in itself, just how practically are they going to get there. And then I suppose we found as well through supported living within (NHS site) our dietician did some work with supported living staff because I think that's a theme that's emerging, sort of paid caring role, there's often a lack of understanding of staff themselves around supporting a healthy diet and healthy portions."* (ID nurse/researcher 01002)

**Participants' enthusiasm for the study.** Despite reservations expressed by some NHS health care staff on the value of the diabetes intervention and family/paid carer engagement with it, most adults with ID/T2D who got to speak with a

researcher about the study were generally enthusiastic about it, demonstrated an acceptable level of understanding and were happy to consent. They were attracted by the possibility of a social group activity, although had some initial concerns in terms of engaging with others in a large group and their ability to understand information given. Like the NHS health and social care staff, they also expressed practical concerns in terms of getting to venues and availability of family/paid carers to take them:

> *"Well, I enjoyed doing the chat with [the recruiter]. Something new for me which is really good."* (Participant with ID S01036)

> *"That's a good idea your boss came up with. It's gonna help people to lose weight and keep more fitter. Like I need to stick to it, that's what I'm saying. I need to find a way to keep myself motivated. I need to find my days to go there, but I'm working on farms and doing other things too with the staff here."* [Participant with ID S01003]

It became clear as the screening and recruitment processes progressed that one of the fundamental requirements for the main study would be to invest time with NHS site teams particularly in the early stages of site set-up to establish relationships, expose views on the study, promote study 'buy-in' from senior managers and front-line staff, and identify anticipated and experienced challenges as recruitment was underway. It also proved important to understand and address potential study participants' and their family/paid carers' concerns pertaining to transport and the time required to support the person with ID/T2D to attend the intervention, to avoid them becoming barriers to participation.

### Theme 2: Discomfort putting potential participants with ID forward for the RCT

**Preconceived judgements on individual suitability for the study.** Eligibility criteria were deliberately kept broad given the pragmatic nature of the study and desire to ensure that adults who had mild/moderate ID with T2D had the opportunity to consider participation. Some NHS health and social care staff stated being unable or unwilling to go through case lists to identify potential participants with ID/T2D for the study. Whilst they frequently cited workload pressures as the reason for this, it became clear that they were uncomfortable with the broad degree of inclusivity at initial screening (particularly those who were less experienced with working with this ID population) and sometimes made subjective and biased judgements as to which participants with ID/T2D should be approached for a recruitment discussion with the researcher. Some had preconceived ideas about who would engage and therefore be suitable for the study. Others had discomfort with putting certain people with ID/T2D forward, even though they met the study eligibility criteria. It was noted that people with ID/T2D who were elderly (for example 63 years), living in a nursing home, in poor health, had no personal transport (despite the 'My Diabetes and Me' study paying for transport to attend the intervention sessions), attended a day centre and, in one case, those who were 'grumpy' were not deemed suitable to be approached about the study:

> *"…most people wouldn't be able to be engaged with that sort of work, it would be too much if they're not managing their Type II diabetes, it's being managed by their support team, you don't want that. Or their disability is too severe and we know that they wouldn't or the support team set up isn't interacting or either they wouldn't be supported to engage with the study or that the family would be suspicious or it would be difficult for them to make it to any sessions for whatever reason."* (NHS clinician 03001).

> *"I think you would only encourage people that you really felt could engage and would really benefit from the programme."* (ID nurse/researcher 01002)

**Consequences of pre-conceived judgments.** As a result of these preconceived views, fewer people than expected were put forward for a study recruitment consultation which frustrated local site researchers as it meant people with ID/T2D missed out on the opportunity of considering trial participation that could benefit their health:

*"…It's about engagement, and people who engage and then disengage with services, and I suppose people are quite benevolent and they're thinking, well, you know,'I don't wanna be wasting the researchers time because I know such and such won't, they'll come for the first session, but then after that they'll disappear off the face of the earth'. So there's that, engagement and compliance and the preconceived ideas are the main issues."* (Researcher R01002)

*"…I know I've seen a few times where they've just written unable to give consent or lacks capacity and literally, lacks capacity. But what does that actually mean? Who said they lack capacity? Does that individual have the right to at least get the information in the post? And then the research team, if they're happy for the research team to make contact, but then at that point could we do a wee bit more in terms of screening?"* (Researcher R01001)

Some NHS health and social care staff did approach people with ID/T2D who they felt would be reluctant to participate and they did participate and were happy to have been given the opportunity to do so. Unintentional gatekeeping, and the consequence of this was highlighted at a study-wide investigator's meeting, drawing on evidence from the QRI to raise awareness and support NHS staff to overcome preconceptions and discomforts with putting people forward for study consideration.

### Theme 3: Emotional labour arising from organisational challenges

**Variations in research governance and difficulties acquiring ID-experienced staff.** The work required behind the scenes in this study in addressing various organisational processes and opinions across different regional jurisdictions, became increasingly apparent as the internal pilot progressed. The labour itself fell mainly on the CI and co-investigators, but centred on the research teams, the NHS health and social care staff and other service staff involved in the data collection, rather than the adults with ID/T2D themselves. Significant challenges arose in relation to site specific organisational processes which the local Principal Investigators (PIs) needed to resolve before they could commence recruitment at their site. This set the study behind schedule and reduced the time and opportunities for recruitment:

*"Unusually, the biggest thing I would say about research and about doing a trial like this is normally the problem used to be ethics. The ethics was actually the easiest thing to get. The biggest problem was getting research governance and there was a thing called C&C which is Capability and Capacity, and each of the three sites are extremely different. There are different ways of working and it's trying to get your head around that."* (Researcher 01007)

One study NHS site in particular experienced a significant delay in the recruitment of staff, which in turn led to delays in participant recruitment. NHS sites also struggled with recruiting clinical research nurses (CRNs) with sufficient experience of working with people with ID, resulting in less experienced staff being recruited than originally planned. Some CRNs, who were experienced with RCT processes but less experienced working with people with ID, had concerns around patient safety, not having information that they felt was necessary for recruiting people to the study, and how the baseline visits were conducted. This explained in part the finding that they were less likely than experienced ID colleagues to be comfortable putting people forward for the study. Adapting preconceived ways of working to fit the needs of the ID population in the present study proved challenging for those who felt limited in their mental health and ID exposure:

*"…I think this study is totally different because we've been told we shouldn't be looking at the records and it's just we've never worked like this before. I think maybe some basic training in mental health, learning disability traits would be quite good. I mean, we sort of touched on it in the education training, but I think maybe something more in depth for the nursing staff would be really good, and maybe scenarios and how to deal with different types of people. I mean they are used to doing research visits in a clinic and following a protocol. It's just we're not used to this group of people."* (Clinical Research Nurse R03001)

These staffing challenges had not been recognised and/or raised prior to the start of the study, so consequently, procedures had to be adapted at short notice, usually resulting in additional work for the co-investigators and CI. This involved much more training and support than had been anticipated.

**Existing organisational pressures exacerbated by the COVID pandemic.** While everyone involved in the study was acutely aware of the pressures within health and social care following the COVID-19 pandemic, it was clear that this had exacerbated existing organisational pressures. All NHS health and social care teams at each site received information on the study protocol and the processes they would be required to undertake, but some of the interviews and minutes of meetings revealed that site staff and the lead research team had underestimated how much time would be required for certain activities in the post COVID climate:

*"…If we could allocate one person in our team to take this on as their baby, yeah, it would work very well but the way things are going with Covid and stuff you never know who's gonna be off and when, so it might be tricky enough, yeah. There hasn't been anything really happening recently but we've been so short in staff, we've had a complete change in staff from management right down over the last few months so….group education and all that, it hasn't really happened."* (NHS Clinician 01003).

**Styles of leadership and challenges maintaining relationships.** In this study, we came across examples of positive leadership, with NHS staff being supported to identify participants with ID/T2D and support the initial recruitment of participants. We also noted examples of fractured leadership, where there was a lack of decision-making and prolonged delays resulting in additional workloads for the researchers. This was in part related to challenges around the relationships within the sites as a result of staff changes following COVID-19. Relationships which had been developed prior to the pandemic had to be revisited as new staff came into post, and in some cases, no-one was currently available, or willing to participate in the same way, leading to further delays as alternative solutions were identified.

This resulted in an increasing responsibility for the co-investigators as they had to expend additional effort finding local partners across the HSC and voluntary sectors who would be prepared to reach out to potential participants:

*"…So, when you have real significant delays around not getting the green light to go, even though the sponsor gave the green light at the start of January. But at the site level if you don't get the green light, we're getting a green light in the three to four months for recruitment, and we know this is a very difficult and hard to reach population therefore you're minimising the maximum you can get out of recruitment."* (Researcher 01007)

## Discussion

Increasingly, there is an aspiration to generate real-world health research, either through routinely collected data or through bespoke data collection [37–38]. The National Institute for Health and Care Excellence (NICE) real-world evidence framework [39] identifies the need to estimate effects of treatment in 'populations excluded from, or under-represented in, the available randomized controlled trials.' This was the first study to use the QRI methodology to identify the enablers and barriers of recruiting participants with ID/T2D to an RCT and the resulting complexity inherent in trials involving this population [40–41]. Previous QRI studies have focused mainly on secondary care services [42–43], and it is clear that even with easier access to potential participants, similar recruitment challenges arose.

### Where do the challenges lie?

Many individuals with ID/T2D, once identified and approached, were open to the study and willing to consent, and this was an ongoing source of motivation and perseverance for all the site teams. Challenges were more prevalent in NHS site

set up and the early stages of the identification and recruitment process, in terms of identifying and approaching potential participants with ID/T2D, as opposed to the later stages of gaining participant consent. Issues were identified with setting up and conducting an RCT with people with ID in a post COVID-19 climate, getting senior and front-line staff buy-in and engagement with a study that supports self-management of health for people with ID, and discomfort putting people with ID forward if staff felt they had limited experience with or had preconceptions of an ID population. As these barriers to recruitment increased, at times seeming to be intractable, the emotional burdens for the research team charged to deliver the study on time and in budget mounted.

### Clear obstacles and hidden challenges

While using the QRI is a novel approach within ID research, many of the issues identified will resonate with researchers in studies outside of the ID world [44]. In some ways, this is reassuring as it illustrates that the challenges of recruitment into RCTs cannot be assumed to solely relate to this ID population itself, which is a common misconception. What the QRI has enabled, however, is the identification of recruitment challenges that were clear to those involved – 'clear obstacles' [45] – namely the organisational and logistical issues to do with staff recruitment and opening delays. But it has also unearthed the 'hidden challenges' [43,45,46] of recruitment such as making judgements on who would be suitable for the study based on preconceptions, which equally hampered recruitment but in a way that was perhaps less obvious to those recruiting. The number of NHS health and social care staff/family and paid carers making judgements on behalf of the ID population is a consistent theme relating to recruitment challenges in research [47–48]. This is an alarming indication of how health and social care professionals are failing to support this ID population to engage in clinical research that could benefit their health [49]. When viewed from an advocacy perspective, it is crucial that people within this population are not taken advantage of. However, from a paternalistic point of view, it excludes people with ID who may be able to make a valuable contribution to research and, demonstrated through their interviews in this study, their pleasure at being included.

### Gatekeeping

There is a significant cohort of NHS health and social care staff, and paid carers, referred to as 'gatekeepers' who have contact with people with ID, which would not be the case in clinical research with less vulnerable populations [50–51]. This creates an additional burden on researchers in relation to communication strategies and emotional labour [3,52,53]. NHS health and social care staff and paid carers acting as gatekeepers can be seen as a protective enabler for the ID community when it comes to ensuring that the ethical standards around capacity and participant wishes are maintained (50), but several layers of gatekeeping can present a systemic barrier that researchers struggle with [48]. Recruiters may have to contend with the intellectual challenge of presenting the study to an individual who fulfills the eligibility criteria, but in whom they feel would be better off with one intervention over another [46,52]. This has the potential to compromise equipoise in their discussion of the study and influence the potential participant's decision making, but can be overcome with appropriate training and support.

### The frustration of organisational barriers

In this study it wasn't so much about how the study was presented to potential participants with ID/T2D or those individuals declining participation due to intervention preferences, but issues much earlier on in the identification and recruitment process. Many of the challenges arose because of insufficient preparation, readiness, and lack of early involvement of the NHS health and social care senior managers and front-line staff. As a result, the work required behind the scenes in this study became increasingly apparent and created an emotional burden. The emotional aspects of working on a clinical trial for those undertaking recruitment has been reported previously [45,47,53], but less so from the angle of those involved with the overall design and delivery, as in the present study. Communication with participants in any research

study is important, but it was clear from these interviews that early and ongoing engagement with both NHS health and social care staff and adults with ID/T2D and their families/paid carers is a core key component to timely and effective RCT recruitment.

## Preparing an RCT submission

There are a number of challenges researchers need to be cognisant of when preparing RCT grant submissions related to any population. Realism and pragmatism must play a big part in any research study, and this should begin with the grant application. There is a tendency to underestimate the budget required for clinical trials and many researchers seek to minimise their request in the hope of a successful grant application [54]. It is also common for only 80% of funding requested to be provided, which may further hamper recruitment efforts later. Research may not be a current priority within NHS health and social care, given the issue of backlogs in care provision partly as a result of the COVID pandemic, which may impact a study's ability to recruit [55] as noted in the present study. As a result, realistic funding must be included to ensure that sufficient high calibre staff are seconded/employed to prioritise research studies, particularly in those populations where additional time is required for recruitment processes, such as the elderly, children, and people with cognitive, intellectual and developmental disabilities. It is also crucial to provide realistic deadlines for study procedures including identification and recruitment. These are often optimistically tight, leading to anxiety and potential conflict when they are not adhered to. Early discussions with study sites to confirm what is achievable are important and recording of decision-making at each stage supports accountability throughout the study.

## Leadership

Support and leadership at every level is essential for a successful trial [56]. This can be difficult to achieve unless researchers engage with NHS organisational leaders at an early stage. Prioritising a research agenda within NHS healthcare has always been challenging [57] and involvement of the relevant organisations in developing the grant submission is one way of ensuring 'buy-in' at an early stage, but researchers should be cautious of assuming that this alsomeans 'buy-in' from NHS health and social care staff, which may not be the case [23]. Early andregular conversations with NHS senior managers and those front-line staff responsible for recruitment are central to the success of any trial. A positive relationship between NHS health and social care staff and research teams can dictate the success of a recruitment strategy, providing clarity of each other's roles and a cohesive approach to recruitment [58]. As we have seen from this study, building, and maintaining relationships is crucial but problematic in NHS health and social care as capacity and capability are increasingly mentioned as barriers to recruitment [59]. This can be an issue for any study, particularly those involving multiple specialties, perceived capability for vulnerable populations or even patient ownership [46,48]. Pre-recruitment work should therefore focus on establishing or renewing those relationships with senior NHS leaders and health and social care staff at each site to address the other obstacles that are an inevitable part of any research study.

## Strengths and limitations

This is the first time the QRI has been incorporated into an RCT involving an ID population with associated health co-morbidities such as T2D. This allowed us to triangulate multiple sources of data to understand how the identification and recruitment processes worked in real-time and where the key bottlenecks were [33]. Although identified issues such as challenges with setting aside personal views when considering study eligibility have been noted previously [52], embedding the QRI in a trial with an ID population enabled us to understand how this specific context shaped previously reported issues, which will be of value to future ID trialists. The QRI was also able to develop and implement actions to address the issues identified, either as the pilot study was underway or to implement in the full study to maximise successful recruitment in this and future trials.

A limitation of this work is the lack of screening log information supplied by two of the study NHS sites which would have helped identify clearer decision-making pathways and unpick any site-specific bottlenecks. Only Site 1, who randomised the majority of participants to the study, provided screening log data. A significant amount of work went into developing the screening logs, but clearly the lack of engagement in returning the information needs to be explored for future studies. We also recognise that we could have included more interviews with participants with ID/T2D who were recruited to the study and perhaps some family carers to provide further insight into the recruitment approach from their perspective. We hope to address this with a further piece of work to capture the views of family/paid carers to strengthen the overall findings from the study. However, we do recognise that the recruitment challenges lay less with the individuals with ID and more with NHS organisational and team working aspects, so it is unlikely that further data from participants themselves would have changed the narrative presented.

### Recommendations

On the basis of findings from the present study, the following implementation recommendations are made to optimise recruitment of adults with ID to randomised trials that support better health:

- Invest time with NHS senior managers and front-line health and social care study site teams in the early stages of study set-up to:

  - establish relationships, expose views on the study such as selection/gatekeeper bias

  - identify training needs for ID nurses such as familiarisation with research processes,

  - promote study buy-in

  - explore anticipated challenges such as identifying how/where potential participants can be approached, e.g., community groups, day centres etc.

  - ensure ongoing monitoring of these issues as recruitment is underway.

- Support NHS sites to collect detailed data on screening and recruitment processes to identify where the core recruitment barriers are and whether they are study wide or site specific. Site staff should be aware of the value of collecting such data and results fed back and discussed to raise awareness and resolve identified issues.

- Highlight specific training issues for CRNs working with the population under study to address potential discomforts such as communication skills on explaining research studies to this population.

- Adults with ID/T2D are generally willing to participate in RCTs to improve their health but have concerns that could act as barriers. Recruiting staff should be supported to elicit and explore their intervention preferences and concerns to support informed decision making on study participation.

- Acknowledge the pressure on the grant holders and particularly the CI, to deliver a funded study on time and in budget despite a plethora of obstacles and barriers that may be beyond their control.

- Advocate for a review of training and support for PIs and co-investigators re roles and responsibilities and problem solving of recruitment challenges.

### Conclusions

Recruiting participants to RCTs from vulnerable or under-served populations, such as people with ID, poses particular challenges beyond those for recruiting less vulnerable populations. A complex multimethod intervention (QRI) to optimise

recruitment integrated in the pilot stage of an RCT to support self-management of T2D in adults with ID has provided insight into the complexities of accessing vulnerable populations and the tension between advocacy and paternalism on the part of HSC/service staff when it comes to recruiting potential study participants. It also unearthed the emotional aspects faced by HSC and research staff undertaking the recruitment and the wider team managing the study. Our findings challenge the prevailing narrative and suggest that the difficulties of undertaking an RCT in an ID population relate more to NHS health and social care staff teams than to people with ID. Section 9 of the UK Research Framework [5] identifies the responsibilities of individuals, organisations and research teams, and within this paper we have highlighted the need to invest time with key stakeholders in the early stages of study set-up to establish relationships, expose staff views on the study and training needs, promote study buy-in, identify anticipated challenges, and for ongoing monitoring of these issues as recruitment is underway.

## Supporting information

**S1 File. COREQ.**
(PDF)

**S2 File. Topic Guide for 1:1 interview with professionals.**
(PDF)

**S3 File. Topic Guide for 1:1 interview with people with ID/T2D.**
(PDF)

**S4 File. Code book.**
(PDF)

## Acknowledgments

The authors would like to thank all those individuals with ID/T2D who agreed to be interviewed for this research study. We are also grateful to all the NHS health and social care and research professionals who set aside time to be interviewed and to submit documentation as part of the data collection process. We also acknowledge all the work of the co-investigators within each study site and the PPI partners who assisted in the development of the RCT itself. We would also like to thank the local and national funding bodies who supported all aspects of this work.

## Author contributions

**Conceptualization:** Rosemary Kelly, Laurence Taggart, Vivien Coates, Maria Truesdale, Alison Dunkley, Nicola Mills.

**Data curation:** Rosemary Kelly, Laurence Taggart.

**Formal analysis:** Rosemary Kelly, Laurence Taggart, Vivien Coates, Nicola Mills.

**Funding acquisition:** Laurence Taggart, Vivien Coates.

**Investigation:** Rosemary Kelly.

**Methodology:** Rosemary Kelly, Laurence Taggart, Vivien Coates, Nicola Mills.

**Project administration:** Rosemary Kelly.

**Resources:** Rosemary Kelly.

**Supervision:** Laurence Taggart, Vivien Coates, Nicola Mills.

**Validation:** Laurence Taggart, Vivien Coates, Nicola Mills.

**Visualization:** Rosemary Kelly, Laurence Taggart, Vivien Coates, Nicola Mills.

**Writing – original draft:** Rosemary Kelly.

**Writing – review & editing:** Rosemary Kelly, Laurence Taggart, Vivien Coates, Maria Truesdale, Alison Dunkley, Michelle Hadjiconstantinou, Kamlesh Khunti, Nicola Mills.

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
