## [Editor Report · Decision Letter 0]

26 Dec 2024

PONE-D-24-54259Who presents the greatest challenge in intellectual disability research - participants or health and research professionals? Findings from the QuinteT Recruitment Intervention within an internal pilot RCT for those with Type 2 DiabetesPLOS ONE?

Dear Dr. Kelly,

Thank you for submitting your manuscript to PLOS ONE. After careful consideration, we feel that it has merit but does not fully meet PLOS ONE’s publication criteria as it currently stands. Therefore, we invite you to submit a revised version of the manuscript that addresses the points raised during the review process.

Please submit your revised manuscript by Feb 09 2025 11:59PM. If you will need more time than this to complete your revisions, please reply to this message or contact the journal office at plosone@plos.org . A rebuttal letter that responds to each point raised by the academic editor and reviewer(s). You should upload this letter as a separate file labeled 'Response to Reviewers'.A marked-up copy of your manuscript that highlights changes made to the original version. You should upload this as a separate file labeled 'Revised Manuscript with Track Changes'.An unmarked version of your revised paper without tracked changes. You should upload this as a separate file labeled 'Manuscript'.

We look forward to receiving your revised manuscript.

Kind regards,

Sarfaraz K. Niazi

Academic Editor

PLOS ONE

Please confirm at this time whether or not your submission contains all raw data required to replicate the results of your study. Authors must share the “minimal data set” for their submission. PLOS defines the minimal data set to consist of the data required to replicate all study findings reported in the article, as well as related metadata and methods (https://journals.plos.org/plosone/s/data-availability#loc-minimal-data-set-definition ).

If your submission does not contain these data, please either upload them as Supporting Information files or deposit them to a stable, public repository and provide us with the relevant URLs, DOIs, or accession numbers. For a list of recommended repositories, please see https://journals.plos.org/plosone/s/recommended-repositories .

Additional Editor Comments

This engaging paper iterates inquiries that align well with the reported findings. However, it could be made more precise by explicitly referencing the study's focus on recruitment challenges in randomized controlled trials (RCTs), such as "Identifying the Key Challenges in Intellectual Disability Research Recruitment: Participants or Professionals?" The long title presented is not appropriate.

Employing the QuinteT Recruitment Intervention (QRI) methodology is innovative and well-suited for identifying recruitment barriers and enablers in vulnerable populations. The multi-method approach—interviews, analysis of recruitment discussions, screening logs, and review of study documents—provides a comprehensive view of the challenges. The thematic analysis is appropriately applied, though the lack of screening log data from two sites is a limitation. Including more perspectives from family or carers could have enriched the findings, as their influence on recruitment decisions is significant.

The conclusions are consistent with the data presented. They highlight the systemic barriers within the healthcare system and among professionals as more significant than participant-related challenges. This finding is insightful and aligns well with the study's objectives. Focusing on organizational buy-in and emotional labor among healthcare professionals and researchers provides a fresh perspective. However, the conclusions could be more actionable by explicitly connecting the identified challenges to broader implications for healthcare research policy and practice.

Citations in the study are appropriate and comprehensive, drawing from a mix of foundational studies and recent advancements. However, some references to key themes, such as advocacy versus paternalism and gatekeeping, could be further developed to deepen the discussion. Additionally, more explicit connections between the study's findings and prior research on RCT challenges in vulnerable populations would strengthen the narrative.

To improve the paper's suitability for publication, the authors could:

1. Refine the title for clarity and precision.

2. Address methodological limitations, such as missing data from screening logs, by discussing their potential impact in greater depth.

3. Discuss how the findings might influence future recruitment strategies or policy adjustments.

4. Ensure a balanced discussion of participant versus professional challenges by integrating more voices from participants or caregivers where possible.

5. Tighten the narrative in the discussion section to link findings to actionable recommendations and broader implications.

6. Incorporating recent literature can enhance your study by providing contemporary insights into recruitment challenges in intellectual disability (ID) research and related fields. Here are some pertinent studies:

a. Shariq, S., Cardoso Pinto, A.M., Budhathoki, S.S. et al. Barriers and facilitators to the recruitment of disabled people to clinical trials: a scoping review. Trials 24, 171 (2023). https://doi.org/10.1186/s13063-023-07142-1. Summary: This review identifies obstacles and enablers in recruiting disabled individuals, including those with intellectual disabilities, into clinical trials. It emphasizes the need for inclusive recruitment strategies to address underrepresentation.

b. Iflaifel, M., Hall, C.L., Green, H.R. et al. Widening participation – recruitment methods in mental health randomised controlled trials: a qualitative study. BMC Med Res Methodol 23, 211 (2023). https://doi.org/10.1186/s12874-023-02032-1. Summary: This study explores perspectives on online and offline recruitment methods to enhance diversity in mental health RCTs, offering insights that could apply to ID research.

c. McCausland D, Haigh M, McCallion P, and McCarron M. IRB challenges in multisite studies: A case report and commentary from the Intellectual Disability Supplement to the Irish Longitudinal Study on Ageing (IDS-TILDA) [version 1; peer review: 3 approved]. HRB Open Res 2024, 7:3 (https://doi.org/10.12688/hrbopenres.13854.1) Summary: This paper discusses ethical review processes in multisite studies involving adults with intellectual disabilities,

d. Jacobsen P, Haddock G, Raphael J, Peak C, Winter R, Berry K. Recruiting and retaining participants in three randomised controlled trials of psychological interventions conducted on acute psychiatric wards: top ten tips for success. BJPsych Open. 2022;8(4):e125. doi:10.1192/bjo.2022.527. Summary: Although focused on psychiatric settings, this article provides practical recruitment and retention strategies that could be adapted for ID research contexts.

e. Jones H, Cipriani A, Barriers and incentives to recruitment in mental health clinical trials. BMJ Ment Health 2019;22:49-50. Summary: This article examines factors influencing recruitment in mental health trials, offering insights that may be relevant to overcoming similar challenges in ID research.

7. Incorporate the one area missing in this manuscript, the recent moves by regulatory agencies towards Real World Evidence (RWE) trials that will be supported by the conclusions drawn in this paper, with discussions on recruitment challenges in clinical research involving vulnerable populations. The FDA’s 2023 Guidance on Diversity Plans to Improve Enrollment of Participants from Underrepresented Racial and Ethnic Populations in Clinical Trials offers insights on planning and implementing inclusive recruitment strategies. While this guidance is not ID-specific, its principles could be adapted for this study's context.

8. Integrate the FDA's emphasis on investigator training for inclusive recruitment and apply it to the context of ID studies.

9. Reference recent literature on recruitment challenges in vulnerable populations (such as the 2023 scoping review on disabled populations) to position the study within the broader clinical research landscape.

10. Include a recommendation to adopt structured recruitment plans that align with FDA diversity guidelines. This could involve pre-trial simulations or mock recruitment to identify bottlenecks early.

11. Conclude by advocating for integrating regulatory frameworks, such as FDA guidance, into future ID research to standardize recruitment practices and improve inclusivity.

12. Discuss the FDA's 2023 Guidance on Diversity Plans in the methods and recommendations sections to underscore the importance of structured recruitment strategies.

13. Cite the FDA’s Guidance for Industry on Clinical Trial Design and Conduct for Pediatric Populations for parallels in designing trials for vulnerable populations.

14. Highlight how the FDA’s focus on inclusivity can complement ethical frameworks like the Declaration of Helsinki in the study's context.

---

## [Author Response · Author response to Decision Letter 1]

6 Feb 2025

1. Formatting has been reviewed and amended.

2. We are exercising caution over release of this data even though it is anonymised due to the fact that we are working with a defined population, both in terms of staff and people with ID/T2D. Data, in terms of deidentified interview transcripts, are available from the corresponding author on reasonable request. This is on the condition that the request fulfils the necessary approvals in place for 'controlled access' data, that participants have agreed to the optional consent to share their anonymised data, and that participant anonymity or privacy is considered not to be compromised.

3. The title has been amended in the online submission.

4. The ethics statement is a sub section of the Materials and Methods section in the manuscript.

5. There was much debate about the title of the manuscript. All the authors came to a consensus that this title was appropriate as it captured the ID/T2D, gatekeeping and QRI elements which we felt all merited equal weight. However, we have reflected on your comments and the authors have agreed to shorten the title and ensure that all of the elements of the paper are captured in the key words to ensure access for as many readers as possible.

6. We agree that including more perspectives from participants/family members would have added weight to the results. However, the quotes chosen from the participants were those which were most clearly expressed during the interviews. In addition, we felt it was important to include quotes which would support the main finding- that the bottlenecks lay with the healthcare professionals and systems, so the paper is more weighted to this. Family members were not included in this study, which we have included as a limitation of the study. We now plan to carry out an additional process evaluation with family/paid carers to obtain their views on the benefits/challenges of the study.

7. We have amended the conclusion section of the manuscript to reflect the current UK Health and Research Policy Framework published by the Health Research Authority in 2023.

8.We have added the following reference to reflect prior research into the issues relating to gatekeeping and engagement:

Earle et al 2020 A critical reflection on accessing women with learning disabilities to participate in research about sensitive subjects through organisaiotnal partnerships. Br J Learn Disabil 48:162-169.

9. Please see our response to point 6.

10. We have added an introductory sentence to the discussion section in relation to real-world applicability. We feel the rest of the discussion section does link to our amended conclusions and stays true to the experiences of everyone involved in the study.

11.Thank you for these helpful references. We have included the reference from Shariq as it refers to people with ID. The other references while very interesting focus more on those with mental health issues which is not the remit of this paper.

12 & 13. We do not feel it is appropriate to use FDA guidance in a manuscript describing a study carried out in the UK. However, your suggestions were helpful in identifying some additional information for inclusion relating to UK guidance. To this end we have included some detail on the guidance contained within the HRA Research in Healthcare Framework and the National Institute for Health and Care Excellence (NICE) real-world evidence framework, which apply across the 4 nations of the UK and therefore directly apply to this study.

14. Please see our response to point 11.

15-19. Please see our response to points 12 & 13.

---

## [Editor Report · Decision Letter 1]

2 Mar 2025

PONE-D-24-54259R1Who presents the greatest challenge in intellectual disability research - participants or health and research professionals?PLOS ONE?

Dear Dr. Kelly,

Thank you for submitting your manuscript to PLOS ONE. After careful consideration, we feel that it has merit but does not fully meet PLOS ONE’s publication criteria as it currently stands. Therefore, we invite you to submit a revised version of the manuscript that addresses the points raised during the review process.

We look forward to receiving your revised manuscript.

Kind regards,

Sarfaraz K. Niazi

Academic Editor

PLOS ONE

Journal Requirements:

**Additional Editor Comments:**

While finding volunteers for randomized controlled trials (RCTs) is never easy, the procedure is even more difficult for those with intellectual disabilities (ID) and Type 2 diabetes (T2D). Using the QuinteT recruiting Intervention (QRI), this study investigated the challenges in recruiting and sought to explain why they arose and how they may be resolved. Three UK National Health Service (NHS) sites participated in the study, which included interviews with individuals with ID/T2D, research experts, and healthcare personnel. The results opened our eyes. Unlike first impressions, many people with ID/T2D were willing to engage; the healthcare professionals were more reluctant. Some employees felt awkward suggesting volunteers since they thought their participation wouldn't be suitable or they wouldn't grasp the study. Others battled the bureaucratic and logistical obstacles inside the NHS, which hampered recruiting prior to it ever starting. One of the most important lessons from this research was that, usually at the organizational level, recruitment difficulties began far sooner than predicted. Some NHS staff members, especially those not experienced in dealing with people with ID, clearly lacked buy-in. Staff turnover and current post-pandemic pressures only aggravate the situation and complicate regular engagement. The research teams had an emotional load as well—always seeking clearances, navigating administrative obstacles, and trying to persuade stakeholders the study was worthwhile. Notwithstanding these difficulties, those people with ID/T2D who were contacted were usually eager and saw the study as a wonderful chance to participate in a social and learning environment. The results ran counter to several medical professionals' presumptions, which held that participants would find it difficult to interact. Gatekeeping primarily played a role in limiting involvement. Determining whether they should even be informed about the study, healthcare providers, support staff, and even family members often made decisions on behalf of individuals with ID/T2D. to seriously consider joining had the opportunity to give joining serious thought since someone else had already made that decision for them. This well-meaning but ultimately limiting approach resulted in hiring fewer people than expected. Fascinatingly, individuals with ID/T2D demonstrated a clear awareness of the study and eagerly participated when given the opportunity. This study emphasizes the need for a change in viewpoint: instead of presuming restrictions, medical professionals and caregivers should concentrate on enabling people with ID to make their own wise decisions. Roadblocks inside organizations also complicated the research. Every NHS location followed its own bureaucratic procedures, and variations in how they handled research permissions resulted in annoying delays. Some sites engaged ID teams early on, streamlining their process, which resulted in more recruiting. Others greatly limited their recruiting capacity by not being able to get the required clearances in time. was yet another challengeedge of intellectual disability presented still another difficulty. Although some clinical research nurses (CRNs) were at ease with RCT procedures, they were reluctant to find volunteers since they had no experience interacting with people with ID. This uncertainty sometimes led to qualified people being turned away, depending more on presumptions than on data. Beyond the mechanics of hiring, this study clarifies the emotional effort required in research with sensitive groups. The study teams had to negotiate NHS staff opposition, control delays, and continually modify their strategy to match the limitations of every site. exacerbating manpower shortages and reducing the ability aggravating manpower shortages and lowering the capacity of healthcare teams to conduct research. Recruitment was either facilitated or hampered in the NHS, mostly depending on the leadership approach used there. Strong leadership and involved teams made sites more successful; poor leadership and high staff turnover presented more serious problems. Despite these challenges, the study offered a comprehensive analysis on how to enhance recruitment. First and most importantly, early interaction with front-line staff and NHS leadership is vital. Before recruiting starts, establishing relationships and getting buy-in can have a big impact. Particularly for those not experienced in dealing with people with ID, training for healthcare personnel can help to lower discomfort and question presumptions. Real-time recruiting data collection and analysis are also crucial since it is difficult to find the actual obstacles without thorough screening logs. Lastly, as recruiting people from underrepresented groups calls for more effort, adaptability, and tenacity than normal RCTs, researchers must fight for enough funding and reasonable deadlines. This study challenges the commonly held belief that recruiting individuals with ID/T2D for clinical research is a daunting task. Rather, the actual challenges come from the hospital system—organizational inefficiencies, gatekeeping, and staff's lack of preparation. The emphasis should be on removing these obstacles going forward so that those with ID have the same chances of being involved in studies as the general population. Better planning, focused instruction, and significant institutional support can help recruitment in this profession become more inclusive and successful. Changing the perspective from exclusion to empowerment will enable individuals with ID the freedom and encouragement needed to make wise research decisions.

---

## [Author Response · Author response to Decision Letter 2]

10 Mar 2025

The following references have been amendened:

9. van Schrojenstein HM, Lantman-de Valk J, Noonan Walsh P. Managing Health Problems in People with Intellectual Disabilities. BMJ. 2008; 337(7683): 1408-1412. doi: 10.1136/bmj.a2507

12. Bishop R, Laugharne R, Shaw N, Russell AM, Goodley D, Banerjee S, et al. The inclusion of adults with intellectual disabilities in health research – challenges, barriers and opportunities: a mixed-method study among stakeholders in England. J Intell Disabil Res. 2024;68(2): 140-149. doi: 10.1111/jir.13097

31. Taggart L, Truesdale M, Carey ME, Martin-Stacey L, Scott J, Bunting B, et al. Pilot feasibility study examining an structured self-management diabetes education program, DESMOND-ID, targeting Hb1Ac in adults with intellectual disabilities. Diabetes Med. 2018;35(1): 137-146. doi: 10.1111/dme.13539

47. Corby D, Sweeney MR. Researchers’ experiences and lessons learned from doing mixed-methods research with a population with intellectual disabilities: Insights from the SOPHIE study. J Intell Disabil. 2019;23(2): 250-265. doi: 10.1177/1744629517747834

53. Meadmore K, Church H, Crane K, Blatch-Jones A, Recio Saucdo AR, Fackrell K. An in-depth exploration of researcher experiences of time and effort involved in health and social care research funding in the UK: The need for changes. PLoS ONE. 2023;18(9): e0291663.

doi: 10.1371/journal.pone.0291663

---

## [Decision Letter · Decision Letter 2]

1 Jul 2025

PONE-D-24-54259R2Who presents the greatest challenge in intellectual disability research - participants or health and research professionals?PLOS ONE?

Dear Dr. Kelly,

Thank you for submitting your manuscript to PLOS ONE. After careful consideration, we feel that it has merit but does not fully meet PLOS ONE’s publication criteria as it currently stands. Therefore, we invite you to submit a revised version of the manuscript that addresses the points raised during the review process.

As you can see, the reviewer principally praises the paper, but recommends some important revisions that you should take into account when revising the manuscript. Importantly please use a reporting guideline (e.g. COREQ as suggested by the reviewer) and submit the filled-in guideline as supplementary file.

We look forward to receiving your revised manuscript.

Kind regards,

Sascha Köpke

Academic Editor

PLOS ONE

Reviewers' comments:

Reviewer's Responses to Questions

**Comments to the Author**

Reviewer #1: (No Response)

2. Is the manuscript technically sound, and do the data support the conclusions?

Reviewer #1: Yes

3. Has the statistical analysis been performed appropriately and rigorously?

Reviewer #1: N/A

4. Have the authors made all data underlying the findings in their manuscript fully available?

Reviewer #1: Yes

5. Is the manuscript presented in an intelligible fashion and written in standard English?

Reviewer #1: Yes

Reviewer #1: (No Response)

**Do you want your identity to be public for this peer review?** For information about this choice, including consent withdrawal, please see our Privacy Policy

Reviewer #1: No

---

## [Author Response · Author response to Decision Letter 3]

31 Jul 2025

1. The topic guide for participants with ID has been added as a Supplementary File (S3)

2. This sample was derived from a group of professionals within a specific field of practice within the United Kingdom. The authors are limited in any additional details they can include to avoid risking the participants' anonymity which was assured as part of the consent process. We have included some general information to illustrate the diversity of roles.

3. We have included a short descriptor of Author 1 who was the interviewer throughout this study.

4. We have recognised the absence of informal caregivers/family members as participants as a limitation of this study. We hope to address this with a further piece of work to capture their views and have stated this in the Strengths and Limitations Section.

5. Details relating to the data collection are included in the COREQ (Supplementary File S1). We have also added some further detail about the interviews in the main manuscript following Table 1.

6. We were not seeking data saturation in this study. We sought to explore the views of people engaged in the internal pilot of the RCT to identify potential enablers and barriers to recruitment and were keen to collect a variety of opinions and experiences to enhance recruitment to the main trial. Participants were not asked to review their transcripts. A meeting which included many of the professionals interviewed, reviewed an early version of the data analysis and provided further information and clarity. The individuals with ID were not asked to review their transcripts as this was felt to be an additional burden for them, but a lay summary of the results has been provided in an easy-read format. This is alluded to on page 23 but following your advice, we have now added a short paragraph on page 10 to clarify this.

7. No software was used. Interviews were transcribed and coded as described in the main manuscript. Member checking with professionals took part during an investigator meeting where several of those who took part in the interviews were present. A further graphic has been included to illustrate how the categories were developed (Fig 1).

8. The UK Framework was added following advice in an earlier review. However, we accept that we should have explored this earlier in the manuscript and have now moved this to the Introduction. We have added some additional thoughts about potential competencies and support for researchers involved in recruitment of vulnerable populations: (a) identifying how/where potential participants can be approached e.g. community groups, day centres etc. (b) Communication skills about explaining research studies (c) Understanding selection/gatekeeper bias (d) Research processes.

9. Braun & Clark, whose thematic analysis framework was used in this work advise avoiding claims of saturation, preferring the use of concepts such as 'information power'. We feel our thematic analysis, in identifying the multiplicity of organisational challenges which can provide learning not just for researchers in the ID field but for all researchers, is an example of how we have met the aims and objectives of this QRI study. As previously stated, we recognise the absence of data from carers as a limitation of this study.

---

## [Decision Letter · Decision Letter 3]

3 Sep 2025

Who presents the greatest challenge in intellectual disability research - participants or health and research professionals?

PONE-D-24-54259R3

Dear Dr. Kelly,

We’re pleased to inform you that your manuscript has been judged scientifically suitable for publication and will be formally accepted for publication once it meets all outstanding technical requirements.

Kind regards,

Sascha Köpke

Academic Editor

PLOS ONE

Additional Editor Comments (optional):

Reviewer #1:

Reviewers' comments:

Reviewer's Responses to Questions

**Comments to the Author**

Reviewer #1: All comments have been addressed

2. Is the manuscript technically sound, and do the data support the conclusions?

Reviewer #1: Yes

3. Has the statistical analysis been performed appropriately and rigorously?

Reviewer #1: N/A

4. Have the authors made all data underlying the findings in their manuscript fully available?

Reviewer #1: Yes

5. Is the manuscript presented in an intelligible fashion and written in standard English?

Reviewer #1: Yes

Reviewer #1: (No Response)

**Do you want your identity to be public for this peer review?** For information about this choice, including consent withdrawal, please see our Privacy Policy

Reviewer #1: No

---

## [Editor Report · Acceptance letter]

PONE-D-24-54259R3

PLOS ONE

Dear Dr. Kelly,

I'm pleased to inform you that your manuscript has been deemed suitable for publication in PLOS ONE. Congratulations! Your manuscript is now being handed over to our production team.

Kind regards,

on behalf of

Professor Sascha Köpke

Academic Editor

PLOS ONE